# Mental Pain Correlates with Mind Wandering, Self-Reflection, and Insight in Individuals with Psychotic Disorders: A Cross-Sectional Study

**DOI:** 10.3390/brainsci13111557

**Published:** 2023-11-07

**Authors:** Alessandro Rodolico, Pierfelice Cutrufelli, Natascia Brondino, Pasquale Caponnetto, Gaetano Catania, Carmen Concerto, Laura Fusar-Poli, Ludovico Mineo, Serena Sturiale, Maria Salvina Signorelli, Antonino Petralia

**Affiliations:** 1Psychiatry Unit, Department of Clinical and Experimental Medicine, University of Catania, Via Santa Sofia 78, 95123 Catania, Italy; alessandro.rodolico@phd.unict.it (A.R.); pierfelicecutrufelli@yahoo.it (P.C.); p.caponnetto@unict.it (P.C.); laura.fusarpoli@unipv.it (L.F.-P.); ludwig.mineo@gmail.com (L.M.); maria.signorelli@unict.it (M.S.S.); petralia@unict.it (A.P.); 2Department of Brain and Behavioral Sciences, University of Pavia, Via Agostino Bassi 21, 27100 Pavia, Italy; natascia.brondino@unipv.it; 3Department of Educational Sciences, Section of Psychology, University of Catania, Via Teatro Greco 84, 95124 Catania, Italy; 4Independent Researcher, 95123 Catania, Italy; gae24@hotmail.it

**Keywords:** psychotic disorders, mental pain, mind wandering, self-reflection, insight

## Abstract

Understanding the cognitive processes that contribute to mental pain in individuals with psychotic disorders is important for refining therapeutic strategies and improving patient outcomes. This study investigated the potential relationship between mental pain, mind wandering, and self-reflection and insight in individuals diagnosed with psychotic disorders. We included individuals diagnosed with a ‘schizophrenia spectrum disorder’ according to DSM-5 criteria. Patients in the study were between 18 and 65 years old, clinically stable, and able to provide informed consent. A total of 34 participants, comprising 25 males and 9 females with an average age of 41.5 years (SD 11.5) were evaluated. The Psychache Scale (PAS), the Mind Wandering Deliberate and Spontaneous Scale (MWDS), and the Self-Reflection and Insight Scale (SRIS) were administered. Statistical analyses involved Spearman’s rho correlations, controlled for potential confounders with partial correlations, and mediation and moderation analyses to understand the indirect effects of MWDS and SRIS on PAS and their potential interplay. Key findings revealed direct correlations between PAS and MWDS and inverse correlations between PAS and SRIS. The mediation effects on the relationship between the predictors and PAS ranged from 9.22% to 49.8%. The largest statistically significant mediation effect was observed with the SRIS-I subscale, suggesting that the self-reflection and insight component may play a role in the impact of mind wandering on mental pain. No evidence was found to suggest that any of the variables could function as relationship moderators for PAS. The results underscore the likely benefits of interventions aimed at reducing mind wandering and enhancing self-reflection in psychotic patients (e.g., metacognitive therapy, mindfulness). Further research will be essential to elucidate the underlying mechanisms.

## 1. Introduction

Psychiatric disorders are conditions characterized by disturbances in thought, emotion, and behavior that impair functioning and reduce the quality of life [1]. Psychotic disorders are characterized by altered perception or interpretation of reality [2]. Core symptoms often consist of hallucinations across various sensory modalities; delusions and disorganized thought patterns, manifesting as incoherent speech or disrupted trains of thought; deficits in insight and judgment are frequently present as well [3]. Concomitant with these “positive” symptoms, individuals may also exhibit “negative” symptoms such as emotional flatness, reduced pleasure, limited speech output, and lack of motivation, which contribute to functional disability [4]. Cognitive impairments, including compromised memory, attention, and executive functioning, further hamper day-to-day activities [5].

Mental pain, mind wandering and self-reflection are dimensions that have been extensively described in the psychiatric and psychological literature, especially in psychosis [6,7,8]. Mental pain also known as “psychological pain” or “psychache”, is a broad term that encapsulates an array of emotional distress and psychological suffering accompanying psychiatric disorders, usually characterized by feelings of guilt, anxiety, terror, isolation, and a sense of overwhelming [9,10,11,12]. It is often found in psychotic disorders [13,14], but it has a pervasive and transdiagnostic nature, beyond the scope of a specific diagnosis [11,15].

Psychache may be linked to mind wandering, a common cognitive phenomenon where the focus drifts from the present moment to internal thoughts, consisting of daydreaming, future planning, or recalling memories [16]; this phenomenon can manifest either deliberately or spontaneously, representing two distinct dimensions [17]. It could be described as a cognitive process that oscillates between two states: an executive control state and a default mode network state, devoted to self-referential thoughts [18,19].

Another dimension that has been extensively described in the psychiatric and psychological literature is self-reflection. This cognitive process, characterized by introspection, involves the deliberate exploration and examination of one’s own thoughts, feelings, and behaviors [20,21], representing one facet of the mentalizing process, specifically the aspect directed inwards [22]. A critical sub-dimension of self-reflection is self-insight, which is the capacity to gain clear, deep, and accurate knowledge about oneself; this process denotes the capacity to critically analyze one’s experiences: through self-reflection, individuals can achieve a deeper understanding of their motivations, emotional responses, and behavioral patterns, thereby fostering self-awareness and self-understanding [23]. Self-insight is pivotal for personal growth and self-improvement, as it enables individuals to identify their strengths and weaknesses, comprehend their reactions to different situations, and make informed decisions about future actions [24,25].

The Research Domain Criteria (RDoC) initiative, spearheaded by the National Institute of Mental Health, underscores the importance of dimensional and transdiagnostic approaches in mental health research, and it advocates for the integration of biological research [26,27]. Our perspective aligns with the RDoC initiative. Indeed, this approach is particularly relevant to the study of mental pain, which exhibits these dimensional, transdiagnostic, and biological characteristics [14,15,28]; the same could be said for mind wandering [29,30,31,32] and self-reflection [33,34]. Mental pain could be aligned with the Negative Valence Systems, serving as a response to perceived threats or losses and engaging neural pathways like the amygdala and the hypothalamic–pituitary–adrenal axis (HPA) axis [35], which are also involved in stress reactions [36]. Self-reflection and insight could fit within the Social Processes domain, particularly under the Perception and Understanding of Self construct, with implicated areas that span from the Posterior Cingulate Cortex, Precuneus, Inferior Frontal Gyrus, and Anterior Insula to the Inferior Parietal Lobule, involving also high-level metacognitive functions and the prefrontal cortex activity, specifically the Ventromedial Prefrontal Cortex [37,38]. Mind wandering could be situated within the Arousal and Regulatory Systems domain of the RDoC framework, specifically under the “arousal” construct, occurring in both low-arousal states, like lack of task engagement, and hyperarousal states, serving as a distraction from stress [39,40,41,42].

Mental pain in psychotic disorders has been the focus of several studies. It has been recognized as a risk factor for suicide, generalizable between different types of populations [13,14,15,43,44,45,46,47,48]. One such study, which involved 25 outpatients with schizophrenia and used Magnetic resonance imaging (MRI) data, found a correlation between high levels of psychological pain, increased suicidal risk, and altered cortical folding gyrification in the fronto-parietal region [14]. Another study with 113 schizophrenia patients identified the Psychache Scale score and alexithymia as independent predictors of attempted suicide, with psychological pain having both direct and indirect effects on suicidality, while alexithymia was only indirectly connected [13]. The association between suicidal risk and psychological pain was also observed in a study using projective technique testing on psychiatric inpatients, which included 101 schizophrenia patients among other diagnoses (71 major depression, 119 bipolar disorder) [49].

Previous literature has suggested the potential role of mind wandering in the severity of psychiatric disorders [50], particularly in patients with depression [51,52], anxiety [52], or attention deficit-hyperactivity disorder (ADHD) [50,53]. Mind wandering has been studied also in experimental settings involving patients with schizophrenia [54,55,56]. These studies found no correlation with positive symptoms and, intriguingly, most of them reported a reduction in mind-wandering activity in comparison with healthy subjects [54,55,56]; nevertheless, one study does not agree with this finding, reporting that people with schizophrenia exhibit a significantly higher mind-wandering frequency relative to controls, stemming from the prevalent notion that psychotic patients often feel detached from reality [57].

Previous research investigated self-reflection as a predictor of well-being [24], and suggested that this ability could be impaired in individuals with psychosis [6,37,58,59]. In schizophrenia, impaired self-reflection has been identified and shown to have a direct correlation with disease insight [59], while mentalization, of which self-reflection is a component, was found to be positively linked with recovery in psychotic patients [60]. Deficits in metacognitive skills [61], alongside specific aspects of anosognosia such as deficits in self-monitoring related to frontal lobe dysfunction [62] and impaired awareness of illness associated with fronto-temporo-parietal asymmetry [63] have been identified in psychotic patients. These neuropsychological findings correlate with symptoms severity and are concurrently linked with diminished self-reflection tendencies [21].

Self-reflection might exacerbate mental pain, since an act of introspection could potentially lead to rumination or dwelling on negative thoughts and feelings [64,65]. Similarly, self-reflection theoretically might contribute to increased mind wandering, as the introspective process frequently involves the mind drifting to past experiences, hypothetical scenarios, and potential future events, a process called “daydreaming” [66].

Psychotic patients often experience a sense of ontological suffering that is often difficult to address through traditional psychiatric symptom measures, traditional scales, and assessment tools [2]. The value of mental pain as a patient-reported outcome (PRO) measure is particularly noteworthy in the treatment of psychotic conditions. It provides a subjective layer of data that can be critical for assessing the efficacy of interventions from the patient’s viewpoint [67]. Although research on metacognitive dimensions has been carried out on psychotic patients, the relationship between outcome measures like mental pain was scarcely studied.

In individuals diagnosed with psychotic disorders, we hypothesized that unpleasant or negative content experienced in mind wandering may exacerbate mental pain. Additionally, we hypothesized that enhanced self-insight could facilitate adaptive coping strategies, thereby mitigating psychache. The study aims to investigate the interrelationship among mental pain, mind wandering, and self-insight in this population. We expect these factors to be interrelated, acting as either mediators or moderators in their association with mental pain. By focusing on the often-neglected dimensions of individual suffering, we seek to provide actionable data for the refinement of therapeutic interventions and patient well-being.

## 2. Materials and Methods

### 2.1. Participants

Patients were recruited by convenience sampling in the “Policlinico G. Rodolico” outpatient clinic (Psychiatry Unit of the University of Catania, Catania, Italy) from September 2022 to April 2023. Patients visiting the clinic for routine check-ups were approached by attending psychiatrists, who introduced the study and assessed initial eligibility based on predefined inclusion and exclusion criteria. Eligible patients were individuals with a clinical diagnosis of a schizophrenia spectrum disorder according to DSM-5 criteria [68]; aged between 18 and 65 years; capability of providing informed consent. Exclusion criteria included substance/medication-induced psychotic disorder; psychotic disorder due to another medical condition; organic brain disorders; intellectual disability; acute psychotic relapse; severe physical illness. All participants provided written informed consent to take part in the study, which received approval from the internal review board (n. 1/2022).

### 2.2. Measures

#### 2.2.1. Mental Pain

We employed the Psychache Scale (PAS) of Holden et al. to assess mental pain [69], which has previously been adapted and employed in the Italian language population and demonstrated good psychometric parameters [46,70]. It consists of 13 items (Likert scales 1 to 5), and it has no sub-factors. This self-report scale allows the identification and measurement of mental pain levels experienced by participants, characterized by intense feelings of negative emotions such as guilt, despair, shame, or loneliness. The PAS aligns closely with the foundational definition of mental pain, and it is the original instrument developed for its measurement. Its widespread acceptance and use in the field further attest to its relevance and robustness [71]. It has shown robust validity, with an effect size of 0.66 for distinguishing between individuals with and without a history of suicide attempts [69]. Additionally, the scale has excellent internal consistency, reflected by a Cronbach’s alpha of 0.92 [69].

#### 2.2.2. Mind Wandering

To evaluate the incidence and nature of mind wandering, we used the Mind Wandering Deliberate and Spontaneous Scale (MWDS) [17,72,73]. This self-report measure, which has been previously validated in the Italian population, showed reliable psychometric properties, including good construct validity as reported by Chiorri et al. [74]. The MWDS differentiates between deliberate (MW-D) and spontaneous (MW-S) mind wandering through its 8 Likert-scale items, loading on two distinct factors. In terms of psychometric reliability, the original validation study reported good internal consistency, with Cronbach’s Alpha values ranging from 0.83 to 0.88 for MW-S and from 0.84 to 0.90 for MW-D [17]. The scale was chosen for its capability to discriminate between spontaneous and deliberate mind wandering, with evidence for concurrent validity with self-reported fidgeting, mindlessness, and other indicators of attentional dysfunction [72].

#### 2.2.3. Self-Reflection and Insight Scale

The Self-Reflection and Insight Scale (SRIS) is a psychological measurement tool to assess an individual’s capacity for self-awareness [20]. Good psychometric parameters were verified when the SRIS was tested in the Italian population [75]. It is a self-report questionnaire that measures two distinct yet related aspects: self-reflection (SRIS-SR) and insight (SRIS-I). The self-reflection subscale refers to the tendency of an individual to engage in introspection. Insight, on the other hand, pertains to the clarity of understanding one’s “inner world”. It consists of 20 items (Likert scales 1 to 6), loading on two factors (SRIS-SR and SRIS-I). We selected this scale due to its original high reliability across both dimensions (SRIS-SR α = 0.91; SRIS-I α = 0.87) [20]. The scale also demonstrated concurrent validity, inversely correlating with depression, anxiety, stress, and alexithymia, while showing positive correlations with cognitive flexibility and self-regulation [20].

### 2.3. Procedures

After securing informed consent, the participants were requested to complete paper forms that encompassed the scales. These questionnaires were conducted in a quiet, private environment to reduce potential distractions. During the data-gathering process, a member of the research team was present to ensure the accuracy and completeness of the information being collected and to help participants to understand their assignments.

### 2.4. Data Analysis

All data analyses were conducted using Jamovi [76], an open source statistical software based on R [77], using its default integrated statistical package where not stated otherwise. Following “Standard 2.3” of the American Educational Research Association (AERA) [78], reliability analyses of the employed scales and sub-scales were completed to assure internal consistency of data, using the psych package [79], calculating and reporting both Cronbach’s Alpha and McDonald’s Omega as suggested by most recent literature [80]. To investigate the relationship between variables, we used Spearman’s rho correlations; partial correlations were also implemented to control for confounding by sociodemographic and clinical variables (occupation, education, marital status, parental status, housing arrangements, years of illness, diagnosis), using the ppcor package [81]. Categorical variables were incorporated as “dummy” variables: occupational status (student/working vs. unemployed), educational level, marital status (married vs. unmarried), parental status (with children vs. without children), housing arrangements (independent housing—living alone/with own family vs. residing with original family), and diagnosis (schizophrenia/schizoaffective vs. other psychosis). Sensitivity analyses (removing each factor at once) and the correlation matrix between confounders and psychometric measures were conducted to evaluate the potential impact of each candidate confounder; correlations amongst these confounders were also examined to confirm their independence. We carried out mediation analyses to understand the indirect effects that the MWDS and SRIS scales and subscales may have on PAS, using the medmod package [82]; we employed nonparametric bootstrapping, creating 5000 resamples. Analyses to explore potential moderation effects were carried out alongside the mediation analyses to ascertain whether the variables interacted to influence each other’s effects, using the same package [82] and the same number of nonparametric bootstrapping. The primary objective of this study was to identify and explore significant associations between variables. As such, while we recognize the importance of power analysis in interpreting null results, the focus of this research was on the elucidation of positive findings. Non-significant results were not a primary focus and, thus, were not extensively explored or interpreted in this context. A *p*-value of less than 0.05 was deemed statistically significant for all the above-mentioned analyses. Bonferroni correction was applied to adjust the significance level for further interpreting correlations robustness between constructs under the most conservative assumption of Type I error occurrence, setting a corrected alpha threshold at 0.0033 (alpha/15). Moderation and mediation analyses did not require Bonferroni correction as these tests are theory-driven and evaluate specific, interdependent relationships among variables. Applying Bonferroni correction in such instances would be overly conservative, increasing the risk of Type II errors without appropriately controlling for Type I errors [83]. In our study, some questionnaires were unfortunately lost by the recruiters. This loss was random and not associated with any specific characteristics of the participants or their responses. Furthermore, it is important to note that we did not discard partially completed questionnaires. They were only discarded if they were not filled out at all, ensuring the integrity of the available questionnaires used in our analysis. Given these circumstances, we believe the data can be treated as Missing Completely at Random (MCAR). As a result, we decided against data imputation to ensure that we did not introduce any potential bias or distortion into our analysis, especially given the limited sample size [84].

## 3. Results

### 3.1. Sociodemographics and Characteristics of the Total Sample

The complete sample, with no missing data, was comprised of a total of 34 participants. This included 25 males and 9 females. The mean age was 41.5 (11.5 SD), ranging from 21 to 63. The mean years of illness were 12.2 (8.81 SD, range 1–36).

Complete characteristics of the total sample are reported in Table 1 (sociodemographics), Table 2 (diagnosis distribution), and Table 3 (Scale measures).

### 3.2. Correlation Analyses

Table 4 shows the correlations (ρ) between measured variables.

Assuming a ρ of 0.80 or above indicates a very strong relationship, values between 0.60 and 0.79 suggest a strong relationship, those between 0.40 and 0.59 imply a moderate relationship, those between 0.20 and 0.39 denote a weak relationship, and coefficients below 0.20 signify a very weak or negligible relationship [87]; these analyses showed the following statistically significant correlations:A moderate direct correlation between PAS-Total and MW-D (ρ = 0.409, *p* = 0.022)A moderate direct correlation between PAS-Total and MW-S (ρ = 0.577, *p* < 0.001)A moderate direct correlation between PAS-Total and MW-Total (ρ = 0.567, *p* < 0.001)A moderate inverse correlation between PAS-Total and SRIS-SR (ρ = −0.553, *p* = 0.001)A strong inverse correlation between PAS-Total and SRIS-I (ρ = −0.753, *p* < 0.001)A moderate inverse correlation between PAS-Total and SRIS-Total (ρ = −0.476, *p* = 0.006)A moderate inverse correlation between MW-S and SRIS-SR (ρ = −0.400, *p* = 0.029)A moderate inverse correlation between MW-S and SRIS-I (ρ = −0.483, *p* = 0.006)A weak inverse correlation between MW-Total and SRIS-SR (ρ = −0.372, *p* = 0.043)A weak inverse correlation between MW-Total and SRIS-I (ρ = −0.363, *p* = 0.044)A weak inverse correlation between MW-Total and SRIS-Total (ρ = −0.282, *p* ≤ 0.001)

Each subscale within every scale demonstrated strong or very strong correlations not only with one another but also with the corresponding total scale. This strengthens the assumption of internal reliability for these scales.

Assuming the most conservative hypothesis of the Bonferroni correction (alpha = 0.0033), most of our findings concerning PAS-Total correlations retained their statistical significance, except for PAS-Total’s correlation with SRIS-Total and PAS-Total’s correlation with MW-D. However, the relationships between MWDS and SRIS lost their significance, barring the total scale’s inverse correlation, which remained significant.

Table 5 presents the partial correlations (*ρ_c_*) between variables, accounting for potential confounders.

The robustness of the correlation’s strength and significance persisted in the partial correlations and throughout all sensitivity analyses.

Some correlations even increased in strength, and all achieved statistical significance, including those correlations that previously did not.

Correlational strength changes are represented in Table 6.

Sensitivity analyses revealed that these changes occurred mostly when age was accounted for(Appendix A).

However, it is worth noting that the significance of the correlations for SRIS-SR and SRIS-Total was lost after applying the Bonferroni correction (alpha = 0.0033). Conversely, SRIS-I correlations maintained their significance after the correction, except for the correlation with MW-Total.

The confounders were not inter-correlated, with a few exceptions (Appendix A). As expected, relationships were observed between marital status, having children, and residing with one’s own family. Interestingly, an inverse moderate correlation was found between having a diagnosis of schizophrenia/schizoaffective disorder and having children.

Most confounders showed no significant correlation with psychometric measures, with the exceptions being education, which had moderate direct correlations with SRIS scales, and the diagnosis of schizophrenia/schizoaffective disorder, which had a moderate inverse correlation with SIRS-SR (Appendix A).

### 3.3. Mediation Analyses

Table 7 shows the indirect and direct effect of predictors (as percentages), for each of the different mediators, on our dependent variable (PAS-Total).

The results indicate that the indirect effects (mediation) of the variables range from 9.22% to 49.8% across different predictors. In the case of MW scales, this range goes from 18% to 49.8%. When SRIS scales are the predictors, the direct effects are overwhelmingly dominant, with percentages ranging from 63.5% to 90.78%. The highest level of mediation is observed when MW-S is the predictor and SRIS-I is the mediator, with nearly half (49.8%) of the effect being indirect. When the predictors were MW-S and MW-Total, SRIS-I’s estimate of the positive indirect effects reached statistical significance; the paths’ betas are shown in Figure 1.

### 3.4. Moderation Analyses

Table 8 reports the estimates for each predictor and moderator in relation to the dependent variable (PAS-Total).

None of the relationships between the predictors and moderators are statistically significant.

## 4. Discussion

### 4.1. Summary of Findings and Interpretation of Results

To the best of our knowledge, this study is the first to investigate the connection between the dimensions of mental pain, mind wandering, and self-reflection in patients with a clinical diagnosis of schizophrenia spectrum disorder. Our analysis revealed that MW scales have a moderate direct correlation with PAS-Total, reaching the range of strong and very strong when accounting for confounders in partial correlations, the strongest being with MW-Total *(ρ_c_* = 0.807). This suggests that amplified mind wandering in subjects with psychosis might trigger mental pain. Conversely, it is plausible to hypothesize that those experiencing higher degrees of mental pain engage more in mind wandering as a form of coping, escaping their distress in this way, partly due to the reduced insight capacity accompanying mental pain—as shown by our mediation analysis. It should be considered that the MW-D subscale had less correlation strength with PAS-Total, suggesting that deliberate mind wandering might act as a buffer, reducing the adverse effects of spontaneous mind wandering on mental pain. This could be attributed to its less harmful impact on self-reflection abilities, as shown by our data, potentially due to its link to more creative and productive thought processes [88,89,90,91]. Accounting for age reduces this difference, an aspect that needs to be further investigated.

In contrast, SRIS scales displayed an inverse correlation with PAS-Total, ranging from moderate to strong, even when accounting for confounders. Specifically, among the SRIS subscales, SRIS-I had the strongest inverse correlation with PAS (ρ = −0.753; *ρ_c_* = −0.735). Our research adds to previous findings [6,58,60,92], suggesting that reflective insight might alleviate mental pain in patients with psychosis, or alternatively, that mental pain might hinder a patient’s genuine understanding of their inner self. Our mediation analysis further highlighted the role of SRIS-I scales as a mediator in the increasing PAS, suggesting a pivotal role in the dynamics of mental pain. Specifically, the most substantial mediation was observed when MW-S was the predictor and SRIS-I was the mediator, indirectly representing 49.8% of the effect. This could indicate that mind wandering might act as a precursor to the disruption of self-reflection processes, with this disturbance then contributing to mental pain by a reduction in effective coping [93]. Alternatively, it may suggest that the disruptive effect of mind wandering on self-reflection processes reflects mental pain itself, suggesting the validity of its assessment as a signal of internal distress [94].

In our data, we observed an inverse correlation between the SRIS and MW scales. This suggests that as individuals engage more in self-reflective activities, they tend to experience a decrease in mind-wandering episodes. Conversely, an increase in mind wandering is associated with a decrease in self-reflective tendencies. This relationship remains consistent even after accounting for potential confounding variables. This could be interpreted as these two cognitive stances occupying antagonistic positions in the mental processes of the subject. While both are distinct components of one’s cognitive landscape, they appear to operate inversely to each other, suggesting a kind of cognitive balance between introspection and unfocused thought. However, our moderation analyses found no evidence that the SRIS or MW scale influenced each other’s relationship with PAS-Total. This suggests that while self-reflection and mind wandering are interconnected processes, they relate independently of each other on mental pain outcome—possibly through the previously described mediation model.

### 4.2. Potential Therapeutic Implications

Our study is designed to offer possible treatment recommendations that have immediate applicability in clinical settings for addressing core elements of psychotic psychopathology. Exploring relationships between our variables might indicate if mind wandering and self-reflection attitudes exacerbate or alleviate mental pain, thus informing potential therapeutic interventions. For instance, if mind wandering serves as a coping mechanism, therapy could guide patients to employ it more effectively; given that mind wandering is part of a broad range of spontaneous thought processes, including creativity and dreaming, its potential role as a resource should not be overlooked [95]. Conversely, if it intensifies mental pain, therapeutic interventions could focus on strategies to manage or reduce mind wandering [96]. It should also be considered that the mediation relationship identified in our study indicates that mind wandering may partially obstruct effective self-reflection. As such, interventions aimed at controlling mind wandering could be a key element in augmenting self-reflection skills, which may in turn alleviate the mental pain experienced by patients. Refining attentional control modules within meta-cognitive therapies could serve as the initial step, paving the way for the subsequent development of self-reflection skills. It is important to note that self-reflection should remain the central therapeutic objective, as it may be the key determinant of the intervention’s ultimate efficacy according to our data [97]. Such therapeutic strategies could focus on teaching methods to direct mind wandering towards more beneficial thoughts or future outcomes. Furthermore, these approaches should prioritize enhancing patients’ capacity for self-reflection, especially insight, as this dimension might in part mediate the effect. Considering this, treatments for patients with psychosis could place an emphasis on cognitive process control, which could enhance this self-reflection component [89,98,99]. Mindfulness-based therapies, which directly engage with these facets, could achieve similar results [100,101,102], albeit with the challenges associated with treating patients with psychosis [103]. Conversely, if mind wandering serves more as a coping mechanism, it might be advantageous to instruct individuals to utilize this capability more efficiently, without intensifying their distress or diminishing their ability for insight, overcoming the dichotomy between mindful and wandering states of consciousness [104]. For instance, individuals might be trained to transition from spontaneous to deliberate mind wandering [105], a tactic that could be more beneficial according to our data. Our study further suggests that an improved capacity to reflect and especially to understand one’s mental processes might aid in better managing and decreasing mental pain. Equally, intense mental pain could hinder an individual’s ability to focus on and comprehend their inner state. Given these findings, it seems wise to advocate for promoting self-reflection and insight as viable therapeutic goals and signs of potential recovery.

### 4.3. Future Research Suggestions

Future research should explore the specific content associated with the negative impact of mind wandering. Understanding the nature of thoughts and images that contribute to distress could be crucial in learning how to navigate and potentially transform this spontaneous process into a healthier one (e.g., mindfulness techniques) [106]. Additionally, it would be worthwhile to explore whether other spontaneous cognitive processes, particularly creative ones, could potentially replace the maladaptive aspects of mind wandering; these processes could channel into more constructive or creative thought, leading to enhanced insight, thereby reducing mental pain [107,108]. Investigations into the role of creativity [109], problem-solving [90,110], meditation [96,111], mindful dreaming [112] and other similar processes in mitigating mental pain would be a significant next step.

The potential role of self-reflection in mitigating mental pain also deserves exploration. Once we establish a correlation, it is essential to delve into the underlying mechanisms. These could be due to the improved coping skills fostered by self-reflection [93] or the possibility that mental pain encumbers clear introspection, potentially leading to a self-perpetuating cycle [25].

### 4.4. Limitations

While self-administered psychometric measures are valid and reliable, they are prone to typical biases and could offer less precision than direct performance measurements, when available [113]. Mind wandering can be experimentally assessed (e.g., electrophysiological methods [31], eye movement analysis [114], and thought probing [115]), and, although empirical measurements correlate well with self-reported outcomes, the correspondence is not flawless [31,32]. Inherently, our study is designed to identify correlations, which means we cannot definitively infer cause–effect relationships, even though the context of available experience strongly suggests them. Despite the small sample size, most findings reached statistical significance. However, this limitation was particularly evident in our mediation analyses, where most indirect effect estimates were not significant, indicating potential power issues. Applying a Bonferroni correction could render some correlational findings not significant; this is not the case for findings related to the PAS scale when confounders are accounted for, except for its correlation with the SRIS-SR scale and the SRIS-Total scale. Notably, this does not involve the SRIS-I subscale, which maintains a significant strong inverse correlation with the PAS scale. Future research should employ a larger sample to robustly ascertain both direct and indirect effects and may benefit from incorporating additional measures for a more comprehensive analysis. The absence of a control group restricted our ability to draw comparisons with other populations, which, while potentially interesting, is not essential to the primary focus of our study. The medication history was considered too intricate to detail for the narrow scope of this study. While exploring its relationship with drug therapy could provide valuable insights into these dimensions, it may serve as a focal point for a future longitudinal study. It should also be considered that our population was predominantly male, reflecting the distribution of diagnoses in our clinical population due to the convenience sampling.

## 5. Conclusions

Our research identified a direct association between mind wandering and mental pain in individuals diagnosed with psychotic conditions. Self-reflection, particularly self-insight, showed an inverse correlation. Mediation analyses revealed that the impact of mind wandering on mental pain may be partly attributed to a deficit in self-insight. This finding suggests that self-insight could be a crucial focus in the cognitive treatment of these conditions. Moderation analyses indicated that the levels of self-insight and mind wandering did not significantly modify each other’s effects on mental pain, implying potential separate interventions for these aspects.

## Figures and Tables

**Figure 1 brainsci-13-01557-f001:**
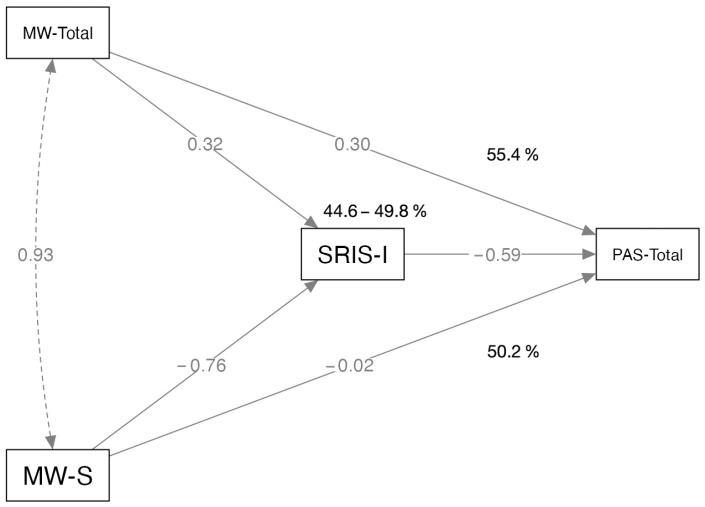
SRIS-I mediation model of MW-Total and MW-S effects on PAS. *β* are indicated on corresponding paths. Relative percentages of direct and indirect effects are also shown.

**Table 1 brainsci-13-01557-t001:** Sociodemographic characteristics.

**Occupation**	*Employed*	5 (14.7%)
	*Students*	4 (11.8%)
	*Unemployed*	25 (73.5%)
**Education**	*Elementary school*	3 (8.8%)
	*Middle school*	16 (47.1%)
	*High school*	13 (38.2%)
	*University degree*	2 (5.9%)
**Marital Status**	*Married*	5 (14.7%)
	*Celibate/Maiden*	27 (79.4%)
	*Divorced*	2 (5.9%)
**Parental Status**	*Has Children*	4 (11.8%)
	*No Children*	30 (88.2%)
**Housing Arrangements**	*With original family*	25 (73.5%)
	*With own family*	7 (20.6%)
	*Living alone*	2 (5.9%)

**Table 2 brainsci-13-01557-t002:** Diagnosis distribution.

DSM-5 Category	N (%)
Schizophrenia	25 (73.5%)
Schizoaffective disorder	5 (14.7%)
Delusional Disorder	1 (2.9%)
* Other Specified Schizophrenia Spectrum and Other Psychotic Disorder	3 (8.8%)

* Patients in this category previously experienced a short-lived psychotic episode, followed by a persistent subclinical state that did not meet definitive criteria for other diagnoses.

**Table 3 brainsci-13-01557-t003:** Scale measures (Items’ Sum).

Scale	Sample Mean (SD)	Internal Reliability	*n*
PAS-Total	26.0 (12.46)	α = 0.940	34
		ω = 0.943	
MW-D	12.8 (5.71)	α = 0.741	31
		ω = 0.758	
MW-S	12.6 (5.90)	α = 0.742	31
		ω = 0.752	
MW-Total	25.4 (10.77)	α = 0.848	31
		ω = 0.853	
SRIS-SR	47.3 (9.12)	α = 0.756	32
		ω = 0.802	
SRIS-I	32.5 (7.92)	α = 0.756	33
		ω = 0.777	
SRIS-Total	78.4 (14.17)	α = 0.755	32
		ω = 0.795	

Given that the alpha and omega values for all scales exceed the commonly assumed threshold of 0.7, we can consider the data collected through these scales as internally consistent for our purposes [85,86].

**Table 4 brainsci-13-01557-t004:** Correlation Matrix.

	PAS-Total	MW-D	MW-S	MW-Total	SRIS-SR	SRIS-I
MW-D	ρ = 0.409 *					
	N: 31					
	*p* = 0.022					
MW-S	ρ = 0.577 *	ρ = 0.734 ***				
	N: 31	N: 31				
	*p* ≤ 0.001	*p* ≤ 0.001				
MW-Total	ρ = 0.567 *	ρ = 0.937 ***	ρ = 0.895 ***			
	N: 31	N: 31	N: 31			
	*p* ≤ 0.001	*p* ≤ 0.001	*p* ≤ 0.001			
SRIS-SR	ρ = −0.553 **	ρ = −0.309	ρ = −0.400 *	ρ = −0.372 *		
	N: 32	N: 30	N: 30	N: 30		
	*p* = 0.001	*p* = 0.097	*p* = 0.029	*p* = 0.043		
SRIS-I	ρ = −0.753 ***	ρ = −0.235	ρ = −0.483 **	ρ = −0.363 *	ρ = 0.632 **	
	N: 33	N: 31	N: 31	N: 31	N: 32	
	*p* ≤ 0.001	*p* = 0.204	*p* = 0.006	*p* = 0.044	*p* ≤ 0.001	
SRIS-Total	ρ = −0.476 *	ρ = −0.276	ρ = −0.328	ρ = −0.282	ρ = 0.896 ***	ρ = 0.624 ***
	N: 32	N: 30	N: 30	N: 30	N: 32	N: 32
	*p* = 0.006	*p* = 0.139	*p* = 0.077	*p* ≤ 0.001	*p* ≤ 0.001	*p* ≤ 0.001

Note: * *p* < 0.05, ** *p* < 0.01, *** *p* < 0.001.

**Table 5 brainsci-13-01557-t005:** Partial Correlation Matrix (controlled for occupation, education, marital status, parental status, housing arrangements, years of illness, diagnosis).

	PAS-Total	MW-D	MW-S	MW-Total	SRIS-SR	SRIS-I
MW-D	*ρ*_c_ = 0.671 ***					
	N: 31					
	*p* ≤ 0.001					
MW-S	*ρ*_c_ = 0.683 ***	*ρ*_c_ = 0.783 ***				
	N: 31	N: 31				
	*p* ≤ 0.001	*p* ≤ 0.001				
MW- Total	*ρ*_c_ = 0.807 *	*ρ*_c_ = 0.944 ***	*ρ*_c_ = 0.912 **			
	N: 31	N: 31	N: 31			
	*p* ≤ 0.001	*p* ≤ 0.001	*p* ≤ 0.001			
SRIS-SR	*ρ*_c_ = −0.505 *	*ρ*_c_ = −0.520 *	*ρ*_c_ = −0.567 **	*ρ*_c_ = −0.584 **		
	N: 32	N: 30	N: 30	N: 30		
	*p* = 0.014	*p* = 0.016	*p* = 0.007	*p* = 0.005		
SRIS-I	*ρ*_c_ = −0.735 ***	*ρ*_c_ = −0.616 **	*ρ*_c_ = −0.659 ***	*ρ*_c_ = −0.712 *	*ρ*_c_ = 0.598 **	
	N: 33	N: 31	N: 31	N: 31	N: 32	
	*p* ≤ 0.001	*p* = 0.002	*p* ≤0.001	*p* = 0.044	*p* = 0.003	
SRIS- Total	*ρ*_c_ = −0.437 *	*ρ*_c_ = −0.577 **	*ρ*_c_ = −0.549	*ρ*_c_ = −0.560	*ρ*_c_ = 0.880 ***	*ρ*_c_ = 0.643 ***
	N: 32	N: 30	N: 30	N: 30	N: 32	N: 32
	*p* = 0.037	*p* = 0.006	*p* = 0.010	*p* = 0.0.008	*p* ≤ 0.001	*p* ≤ 0.001

Note: * *p* < 0.05, ** *p* < 0.01, *** *p* < 0.001.

**Table 6 brainsci-13-01557-t006:** Changes in correlation strength with partial correlations.

Variables	Initial Correlation Strength	ρ Coefficient	Partial Correlation Strength	ρ_c_ Coefficient
MW-D and PAS-Total	Moderate	ρ = 0.409, *p* = 0.022	Strong	ρ_c_ = 0.671, *p* < 0.001
MW-S and PAS-Total	Moderate	ρ = 0.577, *p* < 0.001	Strong	ρ_c_ = 0.683, *p* < 0.001
MW-Total and PAS-Total	Moderate	ρ = 0.567, *p* < 0.001	Very Strong	ρ_c_ = 0.807, *p* < 0.001
MW-D and SRIS-SR	Weak	ρ = −0.309, *p* = 0.097	Moderate	ρ_c_ = −0.520, *p* = 0.016
MW-D and SRIS-I	Weak	ρ = −0.235, *p* = 0.204	Strong	ρ_c_ = −0.616, *p* = 0.002
MW-D and SRIS-Total	Weak	ρ = −0.276, *p* = 0.139	Strong	ρ_c_ = −0.577, *p* = 0.006
MW-S and SRIS-I	Moderate	ρ = −0.483, *p* = 0.006	Strong	ρ_c_ = −0.659, *p* < 0.001
MW-S and SRIS-Total	Weak	ρ = −0.328, *p* = 0.077	Moderate	ρ_c_ = −0.549, *p* = 0.01
MW-Total and SRIS-SR	Weak	ρ = −0.372, *p* = 0.043	Moderate	ρ_c_ = −0.584, *p* = 0.005
MW-Total and SRIS-I	Weak	ρ = −0.363, *p* = 0.044	Strong	ρ_c_ = −0.712, *p* = 0.044
MW-Total and SRIS-Total	Weak	ρ = −0.282, *p* < 0.001	Moderate	ρc = −0.560, *p* = 0.008

**Table 7 brainsci-13-01557-t007:** Effect mediation on PAS-Total.

Predictor	Mediator	Effect	% Mediation	Indirect Effect Estimate [a × b] (SE)
MW-D	SRIS-SR	*Indirect*	33.2%	0.281 (0.184) *p* = 0.126
		*Direct*	66.8%	
	SRIS-I	*Indirect*	38.9%	0.330 (0.225) *p* = 0.142
		*Direct*	61.1%	
	SRIS-Total	*Indirect*	22.9%	0.194 (0.137) *p* = 0.156
		*Direct*	77.1%	
MW-S	SRIS-SR	*Indirect*	28.2%	0.298 (0.193) *p* = 0.123
		*Direct*	71.8%	
	* SRIS-I	*Indirect*	49.8%	0.527 (0.221) *p* = 0.018
		*Direct*	50.2%	
	SRIS-Total	*Indirect*	18%	0.190 (0.136) *p* = 0.160
		*Direct*	82%	
MW-Total	SRIS-SR	*Indirect*	29.3%	0.162 (0.104) *p* = 0.118
		*Direct*	70.7%	
	* SRIS-I	*Indirect*	44.6%	0.248 (0.124) *p* = 0.045
		*Direct*	55.4%	
	SRIS-Total	*Indirect*	19.1%	0.106 (0.0746) *p* = 0.156
		*Direct*	80.9%	
SRIS-SR	MW-D	*Indirect*	16.2%	−0.116 (0.0962) *p* = 0.227
		*Direct*	83.8%	
	MW-S	*Indirect*	29.0%	−0.209 (0.108) *p* = 0.054
		*Direct*	71.0%	
	MW-Total	*Indirect*	26.3%	−0.189 (0.104) *p* = 0.069
		*Direct*	73.7%	
SRIS-I	MW-D	*Indirect*	9.22%	−0.0975 (0.0849) *p* = 0.251
		*Direct*	90.78%	
	MW-S	*Indirect*	17.8%	−0.188 (0.134) *p* = 0.160
		*Direct*	82.2%	
	MW-Total	*Indirect*	15.8%	−0.167 (0.106) *p* = 0.116
		*Direct*	84.2%	
SRIS-Total	MW-D	*Indirect*	21.2%	−0.0793 (0.0610) *p* = 0.193
		*Direct*	78.8%	
	MW-S	*Indirect*	36.5%	−0.137 (0.0782) *p* = 0.080
		*Direct*	63.5%	
	MW-Total	*Indirect*	33.5%	−0.125 (0.0696) *p* = 0.071
		*Direct*	66.5%	

* Mediator whose indirect effect estimate was statistically significant.

**Table 8 brainsci-13-01557-t008:** Effect on PAS-Total (estimates of moderators).

Predictor	Moderator	Estimate (SE)	*p*-Value
MW-D	SRIS-SR	0.0126 (0.0422)	0.765
	SRIS-I	0.00329 (0.0372)	0.930
	SRIS-Total	−0.00115 (0.0306)	0.970
MW-S	SRIS-SR	−0.0212 (0.0422)	0.615
	SRIS-I	−0.00499 (0.0337)	0.882
	SRIS-Total	0.00283 (0.0276)	0.918
MW-Total	SRIS-SR	−0.00588 (0.0218)	0.788
	SRIS-I	−0.00101 (0.0176)	0.954
	SRIS-Total	−0.00198 (0.0148)	0.893
SRIS-SR	MW-D	0.0126 (0.0431)	0.770
	MW-S	−0.0212 (0.0422)	0.615
	MW-Total	−0.00588 (0.0219)	0.788
SRIS-I	MW-D	0.00329 (0.0367)	0.928
	MW-S	−0.00499 (0.0342)	0.884
	MW-Total	−0.00101 (0.0174)	0.954
SRIS-Total	MW-D	−0.00115 (0.0317)	0.971
	MW-S	0.00282 (0.0277)	0.919
	MW-Total	−0.00198 (0.0145)	0.892

## Data Availability

Data are available on motivated request to corresponding author.

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
