# Peer review of "Mental Pain Correlates with Mind Wandering, Self-Reflection, and Insight in Individuals with Psychotic Disorders: A Cross-Sectional Study"

_brainsci, 2023, doi:10.3390/brainsci13111557_

Round 1

Reviewer 1 Report

Comments and Suggestions for Authors

Brief summary: This interesting article explores the potential relationship between mental pain, mind wandering and self-reflection-insight in individuals suffering from a psychotic disorder. The authors included 34 participants between 18 and 65 years old and evaluated them over clinical self-reported scales for the aforementioned elements. They identified that there was a direct correlation between the Psychache Scale and the Mind Wandering Deliberate and Spontaneous Scale. They suggests that there are likely benefits of interventions aimed at reducing mind wandering and there should be relevant attempts to enhace self-reflection in patients suffering from psychotic disorders. Please see my comments below.

Introduction:

- As a general comment, the definition of psychotic disorders as well as broad definition for psychiatric disorder is absent. Since psychotic disorders is the main context for this study, the following structure for the introduction is suggested to account for clarity: Paragraph 1 : Broad definition of psychiatric disorder and a more detailed definition of psychotic disorder. Paragraph 2 : definition of Mental pain, Mind wandering and self-reflection. Paragraph 3: the problematic - what is known in the field and what is lacking. Paragraph 4: The aim and hypothesis of the study.

- Line 63 to 69 must be more elaborated considering this will be relevant for the discussion of the main findings. 

- Are there elements related to self-reflection pertinent to psychotic disorders considering the problem of anosognosia in patients suffering from these conditions?

- Lines 92-96 must be clarified. It is unclear why the ''lack of an outcome'' fully justifies the present study. Your study is relevant and this should be further developed to justify it. 

- Considering the notion of dimensions is discussed in the first paragraph, the RDoC should be discussed way before lines 109-110. 

- The aim was very clear in the abstract but is slightly different in the introduction. What is meant by ''dissect the interplay'' ? 

- Hypothesis should be discussed as well in the last paragraph. 

Material and Methods

- Section 2.1: How were patients recruited / approached? 

- Line 135-136 : The DSM-5 section is Schizophrenia spectrum and Other psychotic Disorders. Were the ''Other psychotic disorders'' involved? For exemple, substance-induced psychotic disorders are currently eligible as per defined by your exclusion criteria. 

- Psychometric properties of all the scales used should be discussed in more details, notably on the level of validity and reliability.

- The choices of scales should also be briefly discussed. 

- It is unclear why 34 participants completed the questionaire and if this number was previously deemed significant on a statistical level. 

- Please explain the choice of not conducting data impuetations (lines 206-207)

Results

- The results are well presented. It is suggested to put lines 268 to 289 in Table form to account for clarity. 

Discussion

- What is being meant by ''the first time'' on line 330?

- While lines 385-287 are true, these therapies already exists. How could existing therapies benefit from your findings? Thise should also be discussed.

- The rest of the discussion is in my opinion very well presented and relevant.

Minor comments:

- To improve the clarity of the manuscript, it is suggested to limit repetition (i.e: Line 42 starts with Mental pain. The next sentence also starts with Mental pain.). 

- Typo: Section 2.4 is missing an ''s'' ''data analysi''

Comments on the Quality of English Language

Nil

Author Response

Comments and Suggestions for Authors

Brief summary: This interesting article explores the potential relationship between mental pain, mind wandering and self-reflection-insight in individuals suffering from a psychotic disorder. The authors included 34 participants between 18 and 65 years old and evaluated them over clinical self-reported scales for the aforementioned elements. They identified that there was a direct correlation between the Psychache Scale and the Mind Wandering Deliberate and Spontaneous Scale. They suggests that there are likely benefits of interventions aimed at reducing mind wandering and there should be relevant attempts to enhace self-reflection in patients suffering from psychotic disorders. Please see my comments below.

Introduction:

- As a general comment, the definition of psychotic disorders as well as broad definition for psychiatric disorder is absent. Since psychotic disorders is the main context for this study, the following structure for the introduction is suggested to account for clarity: Paragraph 1 : Broad definition of psychiatric disorder and a more detailed definition of psychotic disorder. Paragraph 2 : definition of Mental pain, Mind wandering and self-reflection. Paragraph 3: the problematic - what is known in the field and what is lacking. Paragraph 4: The aim and hypothesis of the study.

We thank the reviewer for the suggestions. We modified the introduction following the suggested structure and we included the definitions required. Moreover, as required, we better clarified our hypothesis as it follows:

            “[…] In individuals diagnosed with psychotic disorders, we hypothesized that unpleasant or negative content experienced in mind wandering may exacerbate mental pain. Additionally, we hypothesized that enhanced self-insight could facilitate adaptive coping strategies, thereby mitigating psychache. The study aims to investigate the interrelationship among mental pain, mind wandering, and self-insight in this population. We expect these factors to be interrelated, acting as either mediators or moderators in their association with mental pain. […]”

- Line 63 to 69 must be more elaborated considering this will be relevant for the discussion of the main findings. 

As suggested, we moved the sentences at the end of the introduction to support the hypothesis.

-Are there elements related to self-reflection pertinent to psychotic disorders considering the problem of anosognosia in patients suffering from these conditions?

In psychotic patients’ aspects of anosognosia such as deficits in self-monitoring related to frontal lobe dysfunction and impaired awareness of illness associated with frontotemporoparietal asymmetry have been identified. We enriched the paragraph with a more thorough discussion on relevant literature.

- Lines 92-96 must be clarified. It is unclear why the ''lack of an outcome'' fully justifies the present study. Your study is relevant and this should be further developed to justify it.

We thank the reviewer for appointing this out. We deleted the sentences.

- Considering the notion of dimensions is discussed in the first paragraph, the RDoC should be discussed way before lines 109-110.

We thank the reviewer for appointing this out. We moved the RDoC paragraph to the beginning of the introduction. 

- The aim was very clear in the abstract but is slightly different in the introduction. What is meant by ''dissect the interplay'' ? 

We really thank the reviewer for the statement. We agree with the reviewer. The sentence “dissect interplay” was used to emphasize the need to better understand the differences between dimensions. We acknowledge that the term is unappropriated. We deleted the sentences and rewrote it as it is in the abstract.

- Hypothesis should be discussed as well in the last paragraph. 

 As suggested, the hypotheses are discussed at the end of the introduction.

Material and Methods

- Section 2.1: How were patients recruited / approached? 

Thank you for bringing up the important point regarding patient recruitment. We have revised Section 2.1 to include additional details about the recruitment process. Specifically, we elaborated on how attending psychiatrists approached patients during routine check-ups to introduce the study and assess their initial eligibility. Here it follows the addition sentences to the paragraph:

“[…] Patients visiting the clinic for routine check-ups were approached by attending psychiatrists, who introduced the study and assessed initial eligibility based on predefined inclusion and exclusion criteria. […]

- Line 135-136 : The DSM-5 section is Schizophrenia spectrum and Other psychotic Disorders. Were the ''Other psychotic disorders'' involved? For exemple, substance-induced psychotic disorders are currently eligible as per defined by your exclusion criteria. 

Thank you for pointing out the need to clarify the inclusion and exclusion criteria related to DSM-5 diagnostic categories. We have updated Section 2.1 to explicitly indicate that Substance/Medication-Induced Psychotic Disorder and Psychotic Disorder Due to Another Medical Condition were not included as part of the schizophrenia spectrum disorders in our study. We believe this adjustment resolves the ambiguity and better defines the scope of our research.

            “[…] Eligible patients were individuals with a clinical diagnosis of a schizophrenia spectrum disorder according to DSM-5 criteria (excluding substance/medication-induced psychotic disorder, psychotic disorder due to another medical condition), aged between 18 and 65 years, and capable to provide informed consent.[…]”

- Psychometric properties of all the scales used should be discussed in more details, notably on the level of validity and reliability.

We are grateful for the reviewer's feedback regarding the psychometric properties. To address this, we have incorporated information detailing the reliability and validity of the scales referenced in the manuscript. Our edits underscore the psychometric integrity of these scales, emphasizing their proven robustness. We've elaborated on each scale's internal consistency, validity, and associated statistics. Below are the additions made to each scale's subsection:

2.2.1. Mental Pain

[...]It has shown robust validity, with an effect size of 0.66 for distinguishing between individuals with and without a history of suicide attempts[66]. Additionally, the scale has excellent internal consistency, reflected by a Cronbach's alpha of 0.92 [66]. [...]

2.2.2. Mind Wandering

[...]This self-report measure, which has been previously validated in the Italian population, showed reliable psychometric properties, including good construct validity as reported by Chiorri et al. [70]. In terms of psychometric reliability, the original validation study reported good internal consistency, with Cronbach's alpha values ranging from .83 to .88 for MW-S and from .84 to .90 for MW-D [19]. [...]

2.2.3. Self-Reflection and Insight Scale

[...] Di Fabio’s research indicated a Cronbach’s alpha coefficient exceeding 70%, denoting respectable reliability for the SRIS [71]. Additionally, a commendable concurrent validity was noted when juxtaposed with the Flourishing Scale (FS) and the Resistance to Change Scale (RCS). Specifically, SRIS-IN displayed significant correlations with FS (r=0.37) and a negative correlation with RCS (r=-0.41) [71]. [...]

- The choices of scales should also be briefly discussed. 

Thank you for pointing out the importance of discussing our choices of scales. We provide the following rationale:

Psychache Scale (PAS): [...] The PAS stands out not only because it aligns closely with the foundational definition of mental pain, but also as it is the original instrument developed for its measurement. Its widespread acceptance and use in the field further attest to its relevance and robustness. [...]

Mind Wandering Deliberate and Spontaneous Scale (MWDS): [...] The scale was chosen for its capability to discriminate between spontaneous and deliberate mind wandering, with evidence for concurrent validity with self-reported fidgeting, mindlessness, and other indicators of attentional dysfunction. [...]

Self-Reflection and Insight Scale (SRIS): [...] We selected this scale due to its original high reliability across both dimensions (SRIS-SR α = 0.91; SRIS-I α = 0.87). The scale also demonstrated concurrent validity, inversely correlating with depression, anxiety, stress, and alexithymia, while showing positive correlations with cognitive flexibility and self-regulation. [...]

- It is unclear why 34 participants completed the questionaire and if this number was previously deemed significant on a statistical level. 

Thank you for raising the question about the number of participants who completed the questionnaire in our study. In this study, the primary objective was to identify and explore significant associations between variables. Our focus was on elucidating positive findings, and we did not intend to draw conclusions from non-significant or negative results. This approach is aligned with our exploratory nature of the study. We acknowledge that an a priori power analysis was not performed. However, we believe that our results provide valuable insights and preliminary findings that can pave the way for future, larger-scale research in this area. We updated the data analysis paragraph accordingly, as it follows:

            “[…]The primary objective of this study was to identify and explore significant associations between variables. As such, while we recognize the importance of power analysis in interpreting null results, the focus of this research was on the elucidation of positive findings. Non-significant results were not a primary focus and, thus, were not extensively explored or interpreted in this context.[…]”

Moreover, we acknowledge that the paucity of participants prevents to argue about negative findings, as we had already reported in the limitation section:

“[…]Despite the small sample size, most findings reached statistical significance. However, this limitation was particularly evident in our mediation analyses, where most indirect effect estimates were not significant, indicating potential power issues.[…]”

- Please explain the choice of not conducting data impuetations (lines 206-207)

Thank you for your inquiry. Questionnaires were lost randomly due to mishandling by recruiters. However, only entirely unfilled questionnaires were discarded, preserving the integrity of our data. Given this, and our study's limited sample size, we deemed our data as Missing Completely At Random (MCAR). As a result, we chose against data imputation to avoid introducing potential biases or distorting true variable relationships. Here ti follows the edited paragraph:

            “[…]In our study, some questionnaires were unfortunately lost by the recruiters. This loss was random and not associated with any specific characteristics of the participants or their responses. Furthermore, it is important to note that we did not discard partially completed questionnaires. They were only discarded if they were not filled out at all, ensuring the integrity of the available questionnaires used in our analysis. Given these circumstances, we believe the data can be treated as Missing Completely At Random (MCAR). As a result, we decided against data imputation to ensure that we did not introduce any potential bias or distortion into our analysis, especially given the limited sample size.”

Results

- The results are well presented. It is suggested to put lines 268 to 289 in Table form to account for clarity. 

In response to the reviewer's suggestion, we have converted the text detailing the correlations into a structured format in Table 6. We believe this modification provides a more concise and reader-friendly presentation of the data. Here it follows the new table:

Variables

Initial

Correlation Strength

ρ coefficient

Partial Correlation Strength

ρc  coefficient 

MW-D & PAS-Total

Moderate

ρ = 0.409, p = 0.022

Strong

ρc = 0.671,

p < 0.001

MW-S & PAS-Total

Moderate

ρ = 0.577, p < 0.001

Strong

ρc = 0.683,

p < 0.001

MW-Total & PAS-Total

Moderate

ρ = 0.567, p < 0.001

Very Strong

ρc = 0.807,

p < 0.001

MW-D & SRIS-SR

Weak

ρ = -0.309, p = 0.097

Moderate

ρc = -0.520,

p = 0.016

MW-D & SRIS-I

Weak

ρ = -0.235, p = 0.204

Strong

ρc = -0.616,

p = 0.002

MW-D & SRIS-Total

Weak

ρ = -0.276, p = 0.139

Strong

ρc = -0.577,

p = 0.006

MW-S & SRIS-I

Moderate

ρ = -0.483, p = 0.006

Strong

ρc = -0.659,

p < 0.001

MW-S & SRIS-Total

Weak

ρ = -0.328, p = 0.077

Moderate

ρc = -0.549,

p = 0.01

MW-Total & SRIS-SR

Weak

ρ = -0.372, p = 0.043

Moderate

ρc = -0.584,

p = 0.005

MW-Total & SRIS-I

Weak

ρ = -0.363, p = 0.044

Strong

ρc = -0.712,

p = 0.044

MW-Total & SRIS-Total

Weak

ρ = -0.282, p < 0.001

Moderate

ρc = -0.560,

p = 0.008

Discussion 

- What is being meant by ''the first time'' on line 330?

We apologize for the ambiguous phrasing. What we intended to convey is that this is the first study investigating the connection between the dimensions of mental pain, mind wandering, and self-reflection in patients with a clinical diagnosis of schizophrenia spectrum disorder. We have revised the sentence for clarity and appreciate the reviewer's attention to this detail.

            “This study is the first to investigate the connection between the dimensions of mental pain, mind wandering, and self-reflection in patients with a clinical diagnosis of schizophrenia spectrum disorder, to the best of our knowledge.[…]”

- While lines 385-287 are true, these therapies already exists. How could existing therapies benefit from your findings? Thise should also be discussed.

Thank you for pointing out the need to clarify how our findings can be integrated with existing therapeutic methods. We recognize the existence of cognitive and meta-cognitive therapies and have sought to highlight the unique nuances our research brings to the table, particularly in how mind wandering and self-reflection interplay in the context of schizophrenia spectrum disorder.

In light of your feedback, we have expanded upon the implications of our findings within the context of the existing therapies:

"[…] It should also be considered that the mediation relationship identified in our study indicates that mind wandering may partially obstruct effective self-reflection. As such, interventions aimed at controlling mind wandering could be a key element in augmenting self-reflection skills, which may in turn alleviate the mental pain experienced by patients. Refining attentional control modules within meta-cognitive therapies could serve as the initial step, paving the way for subsequent development of self-reflection skills. It's important to note that self-reflection should remain the central therapeutic objective, as it may be the key determinant of the intervention's ultimate efficacy according to our data. In particular, development of insight should accompany the mere tendency to self-reflect, because that is the principal factor that could have a positive effect on the outcome. […]"

- The rest of the discussion is in my opinion very well presented and relevant.

Minor comments:

- To improve the clarity of the manuscript, it is suggested to limit repetition (i.e: Line 42 starts with Mental pain. The next sentence also starts with Mental pain.). 

Thank you for bringing this to our attention. We have reviewed the introduction and made the necessary revisions to reduce repetitions. This restructuring was done in accordance with your earlier recommendation to improve the clarity of the manuscript.

- Typo: Section 2.4 is missing an ''s'' ''data analysi''

Thank you, we corrected the typo.

Reviewer 2 Report

Comments and Suggestions for Authors

A very nice piece regarding Mental Pain correlates with Mind Wandering and Self Reflection in Psychotic disturbances.

Paper is of interest for the readers.

You should add the ethical approval number also in the Methods section, not just at the end

the is an "i" missing in the data analysis subchaper title

introduction sets the scene very very well

Results are definetelly interesting for people working this area, as well as related areas

Discussion section is very well constructed.

A fair chapter of limitations.

Also, conclusions can and should be reduced to only the important matters the authors did observe in their nice study. the other general part don t belong here, and can be moved to Discussion section

Author Response

A very nice piece regarding Mental Pain correlates with Mind Wandering and Self Reflection in Psychotic disturbances. Paper is of interest for the readers.

-You should add the ethical approval number also in the Methods section, not just at the end

Thank you for bringing this to our attention. We have now added the ethical approval number in the Methods section as suggested.

-the is an "i" missing in the data analysis subchaper title

Thank you for pointing out the typographical error in the subchapter title; it has been corrected.

-introduction sets the scene very very well

We appreciate your positive feedback on the Introduction

Results are definetelly interesting for people working this area, as well as related areas

Discussion section is very well constructed. A fair chapter of limitations.

We thank the reviewer for the comments.  

Also, conclusions can and should be reduced to only the important matters the authors did observe in their nice study. the other general part don t belong here, and can be moved to Discussion section

Thank you for your valuable feedback on the conclusion section. We have revised the conclusion to be more concise and focused on our main findings:

"Our research identified a direct association between mind wandering and mental pain in individuals diagnosed with psychotic conditions. Self-reflection, particularly self-insight, showed an inverse correlation. Mediation analyses revealed that the impact of mind wandering on mental pain may be partly attributed to a deficit in self-insight. This finding suggests that self-insight could be a crucial focus in the cognitive treatment of these conditions. Moderation analyses indicated that the levels of self-insight and mind wandering did not significantly modify each other's effects on mental pain, implying potential separate interventions for these aspects."

Regarding the parts that were removed from the conclusion, we opted not to move them to the discussion as they were already covered in that section.